# Nature's Contributions to People Shape Sense of Place in the Coffee Cultural Landscape of Colombia

**Beatriz E. Murillo-López** [1,*] , **Antonio J. Castro** [2,3] **and Alexander Feijoo-Martínez** [1]

1   Faculty of Environmental Sciences, Technological University of Pereira, Pereira 660003, Colombia; afeijoo@utp.edu.co
2   Andalusian Centre for the Assessment and Monitoring of Global Change (CAESCG), Department of Biology and Geology, University of Almeria, 04120 Almeria, Spain; acastro@ual.es
3   Department of Biological Sciences, Idaho State University, Pocatello, ID 83209, USA
*   Correspondence: betymu@utp.edu.co; Tel.: +57-30-0652-3575

**Abstract:** Understanding the roots of a sense of place in farmlands is crucial for stopping rural exodus to urban areas. Farmers' experiences related to their way of life, peace and quiet, rootedness, pleasure, and inspiration are fundamental components of a sense of place in farmlands. Here, we used the city of Pereira located in the Coffee Cultural Landscape of Colombia (CCLC) to examine the role of nature's contributions to people (NCP) in forming meanings and attachments that shape their sense of place to this region. This region has experienced intense agricultural lands abandonment due to rapid urbanization over the last decades. To do so, a mixture of qualitative and quantitative methods was used, including semi-structured interviews, observation, and dialogue, to capture farmers' perceptions and emotions associated with farmlands, reasons for remaining, and the diversity of NCPs. Results indicated that farmers recognized farmlands as a quiet and safe space that support family cohesion. Results also showed that the characteristics of the farms (e.g., agricultural practices, distance to cities, and gender) play an important role in articulating a farmer's attachment to farmlands. Finally, farmers identified nonmaterial NCP (e.g., physical and psychological experiences and supportive identities) to be the most important contributions for shaping their sense of place. We call for the need to include robust and transparent deliberative and negotiation mechanisms that are inclusive of all relevant stakeholders, to aim to address unequal power, and to recognize and strengthen communities' mechanisms of action on the CCLC.

**Keywords:** socioecological systems; local identity; rural abandonment; agroecology; world heritage site

## 1. Introduction

According to the latest report of the Intergovernmental Science-Policy Platform on Biodiversity and Ecosystem Services (IPBES), the supply of food, energy, and materials to human communities is increasing at the expense of nature's capacity to provide, producing drastic effects on ecosystems that sustain livelihoods [1]. In the processes of human use and modification of nature's resources, relationships between people and lands are formed and evolve over time, shaping cultural roots to the land. Understanding this human–nature relationship requires approaches that capture factors that articulate a sense of place, including meaning, attachment, characteristics of places, the complexity of environmental values, and individual experiences within the landscape [2].

The transformation of the ecosystems in the central Andes of South America has configured in the Colombian coffee-growing region environments in which the cultivation of diverse varieties of coffee has predominated, which have given rise to exports to international markets [3–5]. Traditional coffee crops are accompanied by multiple subsystems that form mosaics and patches between successions of natural vegetation, riparian areas close to bodies of water, *Guadua angustifolia* and the predominance of cultivated plants as companions of the systems, which are friendly to the conservation of the biodiversity of the

macrofauna of the soil [6]. However, the intense use and transformation of the traditional farming and natural system (gallery and/or riparian forest and bamboo forest) in favor of urban expansion (discontinued urban fabric) is producing a decline of traditional farmlands systems (traditional coffee and plantain crops) and their biodiversity [7], thus altering the sustainable way of living of rural communities [8].

This context of the Coffee Cultural Landscape of Colombia (CCLC) led to its declaration in 2011 as a World Heritage Site by the United Nations Educational, Scientific and Cultural Organization (UNESCO). The CCLC is considered a landscape that should be prioritized for preservation because of its tangible and intangible significance to the territory, and it is at risk of losing its unique sociocultural roots that rural families have formed with traditional farming systems present there [9]. Among the major risks are urban expansion (e.g., construction of condominiums increases the discontinuous urban fabric) and the intensification of the agriculture (e.g., cattle pastures and plantain and avocado monocultures), which have caused a simplification and homogenization of the landscape, displacing agricultural lands with traditional uses and their communities, leading to the loss of agricultural culture, biodiversity, and sense of place [9]. Together, these land transformations have particularly changed the agricultural practices of the city of Pereira located in the western foothills of the Cordillera Central above the Cauca River valley.

The most dominant farming practices in the CCLC are peasant and semi-industrial styles. The semi-industrial style centralizes labor productivity and growth, mainly based on the mobilization of external resources, which leads to a disconnection between traditional farming and nature, while the peasant style focuses on autonomy, family labor, and self-controlled resources that depend on the sustainable use of ecological capital [10,11]. These farming styles highlight the different ways in which farmers relate to farm resources and production as a business, as well as provide care for families [11,12]. These farming styles also differ in the environmental pressure they place on ecosystems and in the diversity of nature´s contributions (NCP) they provide to people [13].

This investigation adds to the growing body of research addressing the connection between people and nature through the assessment of how NCP shape a sense of place in rural settings (Figure 1). Our intention is to understand the relationship between farming style, sense of place, and NCP, because these concepts are solidly rooted in cultural repertoires. To advance this aim, it is necessary to not only recognize and integrate the characteristics of farmers and farms, but also to explore rootedness, security, and feelings associated with farmlands [14]. By addressing these factors, it will be possible to provide a better and more informed guidance in the future on sustainable land management in these areas [15].

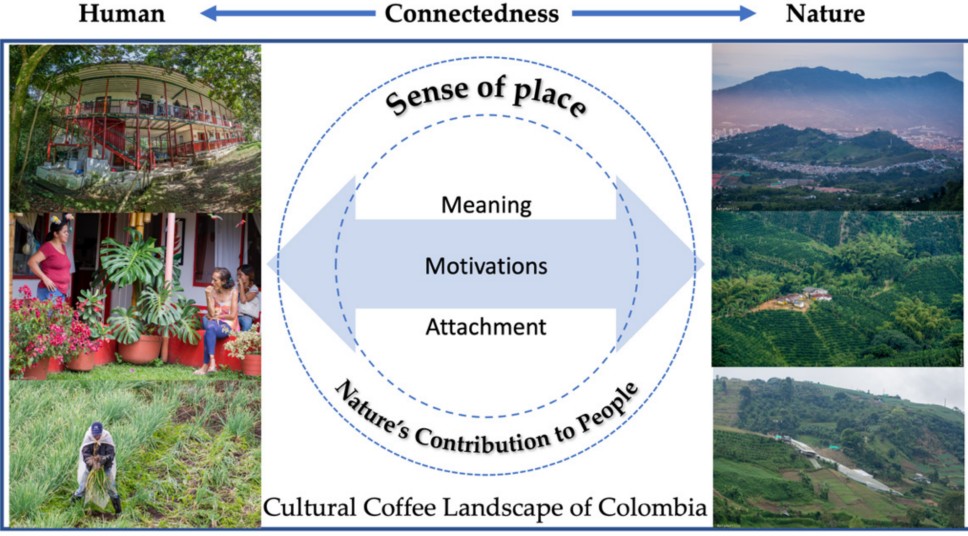

**Figure 1.** Conceptual framework of a sense of place through the NCP lens.

Sense of place is defined as a motivation for stewardship and actions to care for the environment and use the resources it provides. It is also presented as a cognitive and emotional variable that mediates how people respond to social-ecological change [16,17]. The human–nature relationship is nonlinear and often depends on the formations of relational values, i.e., values that arise from a relationship with nature, encompassing a sense of place, feelings of well-being (mental and physical health), and cultural, community, or personal identities [18–21]. Farmers have a complex relationship with farmlands as they have the ability to read nature and make decisions to protect or use resources. Additionally, farms are multifunctional landscapes (e.g., areas production, conservation, and relaxing zone) that can be related to specific relational values of farmlands [12,22,23]. The CCLC is shaped by mosaics (e.g., patches of interconnected crops and natural areas) and are inhabited by rural families holding beliefs, attitudes, and social norms that create farmland with high cultural value. Sense of place in this region has been described as a wide range of connections between people and places that develop based on the place meanings and attachment a person has for a particular setting [16,24]. We integrated the concept of nature's contributions to people (NCP) framework developed by IPBES to capture a broad range of worldviews, knowledge systems, and stakeholders. The NCP approach recognizes the central and pervasive role that culture plays in defining all links between people and nature [21], and the importance of local knowledge for understanding meanings, motivations, and attachment to agricultural landscapes (Figure 1).

Within this context, this study aims to examine the role of NCP in shaping the sense of place of farmers in the CCLC. Specifically, we focused on examining the role of meanings, attachments, values, and connection associated with nature in shaping the sense of place to this region. To do so, a mixture of qualitative and quantitative methods were used to (i) characterize the diversity of farmers and farms of a case study located in the CCLC; (ii) examine the diversity of emotions associated with farmlands, as well as sociodemographic factors that explain them; (iii) explore the sense of place of local communities through exploring motivations to remaining in the region; (iv) identify the diversity of nature's contributions to people that articulated farmers´ sense of place; (v) to explore the visions of local communities regarding the future of the CCLC.

## 2. Materials and Methods

### 2.1. Study Area: The Coffee Cultural Landscape of Colombia

The study was conducted in the rural area of the city of Pereira, Risaralda, Colombia, located between 4°43′4.8″ N and 75°50′38.4″ W and 4°52′15.6″ N and 75°36′18″ W. The farms are located between 1221 and 1922 m.a.s.l. (meters above sea level) (Figure 2). The average temperature is 21.2 °C; the average total annual rainfall was 2301 mm and the relative air humidity ranges yearly between 73 and 79% [25]. Pereira occupies an area of 607 km$^2$ and the approximate population is 467,269 inhabitants, of which 81,432 (17.4%) are residents of the rural area [26].

#### 2.1.1. Land-Use and Land-Cover Change in the CCLC

Over the last three decades, significant changes in land use and cover have been documented in the CCLC affecting the agricultural production of coffee and other native crops. In 1997 the export in Colombia of agricultural products was 32.5% of the total exported; however, in 2011 it was reduced to 8.2% of the export of agricultural products [27]. Changes in land cover and urban expansion in the city of Pereira begin to show the decrease in lands used for coffee cultivation (from 1997 to 2014 it went from 10,706 ha to 5454 ha). Likewise, permanent crops decreased from 5747 ha to 3646 ha for the same period of analysis and transitory crops decreased by 214 ha [9], which placed more pressure in the rural sector due to the change in the type of agricultural production (i.e., pastures for cattle, industrial avocado cultivation) and livelihood of rural communities (i.e., land for human occupation—gentrification), thus influencing factors that shape the sense of place, identity, and heritage.

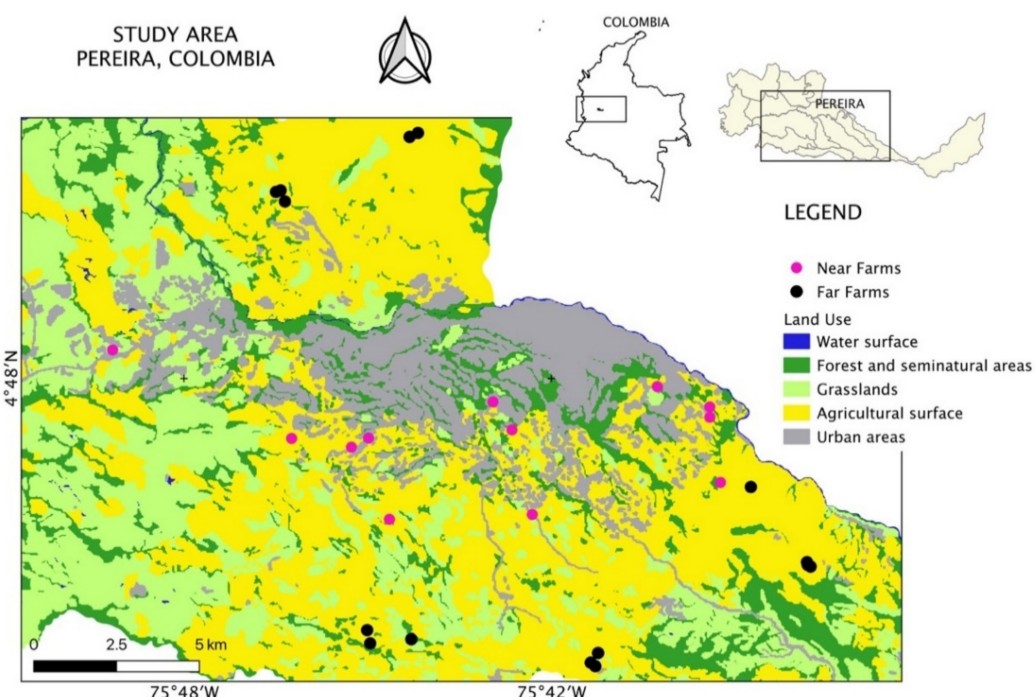

**Figure 2.** Geographic location of farms in the CCLC and current land cover and land-use type.

2.1.2. Farms Characteristics and Locations in the CCLC

Pereira is a municipality of the CCLC and extends through some of the coffee-producing areas at the foothills of the western and central mountain ranges of the Cordillera de los Andes. The characteristics of the area reflect the process of adaptation of coffee cultivation to the complex conditions imposed by the Colombian Andes [28]. The CCLC represents traditional forms of human settlements with small-to-medium-sized production units (between 0.5–2.6 ha), with steep slopes (15–50% inclination), elevation between 1000–2000 m.a.s.l., precipitation between 1600–2700 mm, and average temperature of 22.2 °C [9,29,30].

The CCLC is a continuously productive landscape that has shaped the cultural connection of rural communities to the land over decades. The coffee-growing families have mainly planted coffee, accompanied by subsistence crops (corn, beans, plantain, fruit trees) and with a low level of mechanization. The cultural practices have been passed down through generations and reflect a knowledge based on experience and understanding of the surroundings [8]. In addition, this small-scale production is distinguished by its family-based workforces, whereby the producer and family all work on the farm. Most families tend to live on the premises and so are able to constantly supervise their coffee plants and other crops. Only when the production cycle is at a peak are workers from outside the family hired—on a temporary basis—to help with harvesting [28,30,31]. The farm work is often built on the family farm by doing, making mistakes, correcting them by repeatedly reperforming the activities, and by observing and hearing experiences of neighboring farmers [8]. Farms are centers of (informal) education for families, mainly about crops, practices, and strategies, making the families and their farms into an expression of coffee culture.

Exploring the sense of place in the CCLC requires methodologies that can reveal meanings, attachments, relational, and historical values to these lands [32]. We selected 27 farms based on their proximity to agricultural areas of Pereira (i.e., no forest and seminatural areas, no artificial surfaces), primary productive activity (i.e., no livestock, no tourism), and farmer willingness to participate in this study (Figure 2).

### 2.2. Social Sampling Strategy

Farms were selected by the willingness and desire of rural families to provide information on the values and perceptions they hold in relation to farms and rural landscapes. This study conducted a qualitative research method through the use of semistructured interviews, in-person observation, and informal conversation with farmers [33,34] (Figure 3). The strength of these techniques lies in the creation of bonds of trust between farmers and interviewers to obtain information reflecting meanings and attachment to farmers' values related to the agricultural landscape. Since farmers are often heterogeneous in terms of their relationship with the environment, it was crucial to develop a relationship of trust. This method has been previously used to collect information about emotional connections to natural features and rural landscapes [8]. A total of 27 in-person interviews across all selected farms were conducted between August and December 2018.

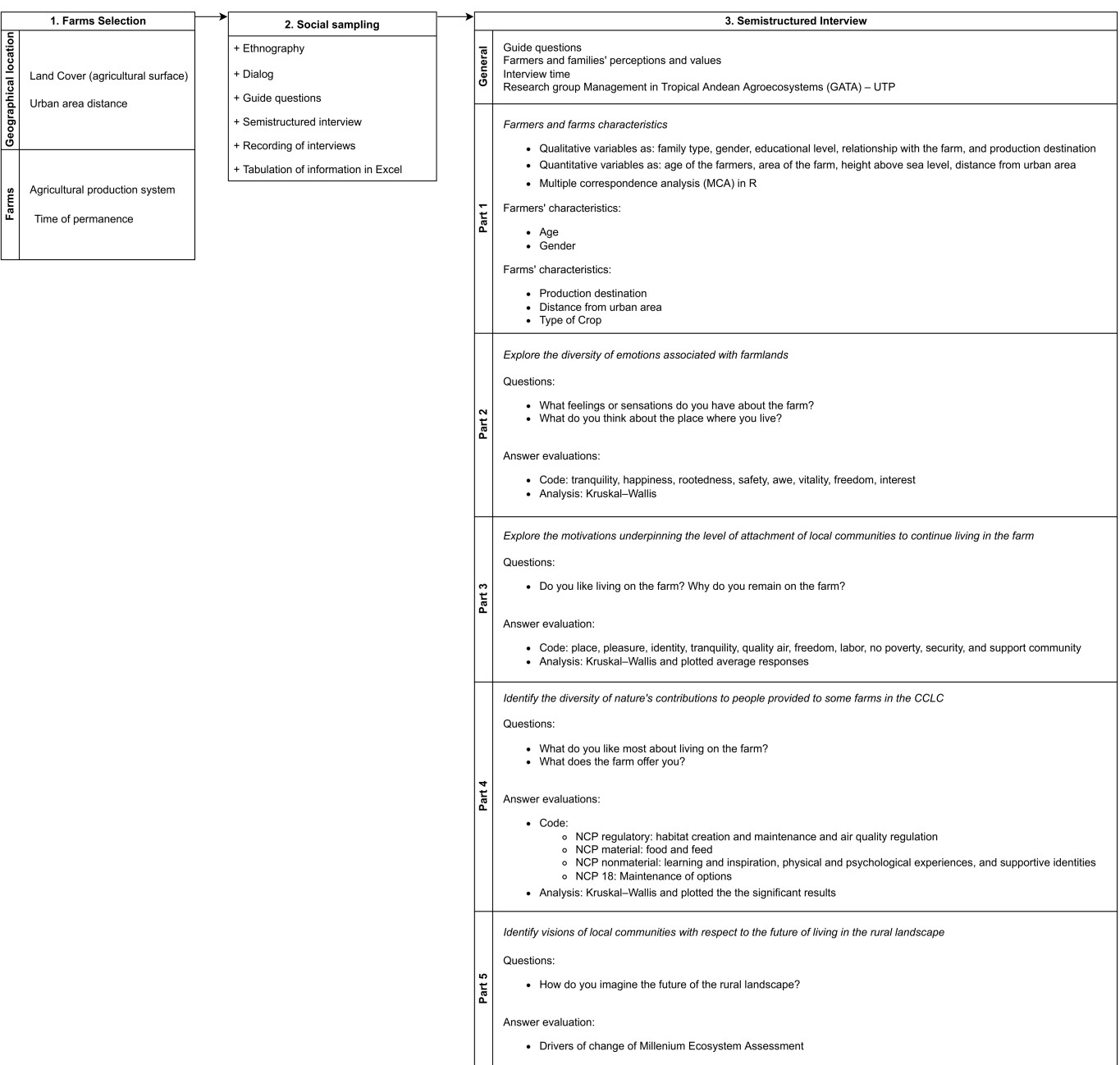

**Figure 3.** Methodological steps of the research approach.

*2.3. Semistructured Questionnaire Design*

The semistructured questionnaire was separated into five sections (Figure 3). In-person interviews were on average one and a half hours and were conducted by the research group Management in Tropical Andean Agroecosystems (GATA, Spanish acronym)—Technological University of Pereira (UTP). The semistructured interviews included open questions aiming to explore farmers and farm characteristics, as well as their perceptions and feelings associated to rural landscapes. In addition, farmer's motivations to remain in the farm and the diverse contributions (i.e., NCP) they perceived from the rural landscapes were explored. Once permission was obtained from the interviewees, each interview was recorded to facilitate the information collection of the interviewee's story and keep the details exactly as they were expressed [35].

2.3.1. Farmers and Farms' Characteristics

The questionnaire collected qualitative information such as family type (childless couples, nuclear, extended) [8], gender (female and male), educational level (primary school, high school, technical, technology, university degree), origin, type of relationship with farms (managers, owner-managers, owners, workers) and destination of crop production (sale and self-consumption, sale). Additionally, quantitative information related to the farmers' age, farm surface area, altitude, time of tenure and time spent on the farm were collected [7]. Farmers and the 27 farms' characteristics were classified based on the data provided by the interviewees and farms' information. We chose two ranges for each qualitative variable; the range was calculated by subtracting the minimum value from the maximum value of the data set, and this range was divided by two to classify farmers and farms according to the characteristics of the group (Figure 3).

A farming style is defined as a distinctive way of ordering the many sociomaterial interrelations involved in farming [11]. Each farming style is a description of the way farmers and rural families arrange the available resources (e.g., labor, land, input, and time) for the exploitation and replication of the production system [10,11,22,36–39]. Information collected from each farm was used to classify them as peasant or semi-industrial style. We used variables such as farmer's relationship to the farm and time living in the farm as well as farm surface, crop types, and which crops generate income; information related to the tenure of the farm, hiring personnel, and destination of the production were taken into account.

Farms were classified as near or far from Pereira City. To determine the distance (near or far), a layer of roads of the municipality was assembled [40] and a distance matrix was created. The type of road was taken into account (levels of difficulty according to the conditions of the roads—earthen roads to cement concrete road—where the value ranged from 1 to 7, with 7 being the weight of the road with the greatest difficulty to be traveled by farmers to carry agricultural production to the city). The matrix was generated from the farms to the market place in Pereira. The result was a matrix with the weight of the roads (distance in meters and the value in difficulty of the roads to reach the center of Pereira) (Table A1). To analyze the characteristics of the farmers and farms, a multivariate analysis was performed using the age and gender of farmers and the distance to the urban area, type of crops, production destination, and area. A multiple correspondence analysis (MCA) in R was used to explain the relationship between types of farmers and farms' characteristics.

2.3.2. Farmer's Emotions Associated with Farm Landscapes in the CCLC

We asked farmers about their emotions generated by living on these farm landscapes. We introduced different questions to explore their perceptions and facilitate the dialogue with farmers. The following questions were asked: What feelings or sensations do you have about the farm? What do you think about the place where you live? Responses were coded according to eight emotions associated with living on the farm, including tranquility, happiness, rootedness, safety, awe, vitality, freedom, and interest (see Tables A1 and A2). Several emotions could be associated with one single response. A Kruskal–Wallis analysis

was performed to find correlations between farmers' and farms' characteristics and the diversity of emotions (Figure 3).

### 2.3.3. Sense of Place of Local Communities in the CCLC

The sense of place within farm landscapes was examined by using multiple questions, including Do you like living on the farm? Why do you remain on the farm? Responses were coded according to motivations to continue living on the farm and classified as place, pleasure, identity, tranquility, air quality, freedom, labor, no poverty, security, and support community (see Tables A1 and A2). The Kruskal–Wallis analysis is a nonparametric test for comparing variances of more than two variables and it was used to explore differences between farmers' and farms' characteristics with motivations to continue living in the CCLC (Figure 3).

### 2.3.4. Diversity of Nature's Contributions to People Provided by Farms in the CCLC

To explore the diversity of NCP associated with farms, the following questions were asked, including What do you like most about living on your farm? What does the farm offer you? Each response was transcribed and classified into the material and nonmaterial NCP proposed by Díaz et al. [21]. Considering the mean of the responses, a Kruskal–Wallis analysis was performed to explore the relationship between farmers' and farms' characteristics and NCP (Figure 3). NCP were grouped into material, nonmaterial, and regulating categories. In these categories NCP18 was not included because this contribution is considered in the three groups (material, nonmaterial and regulating NCP) for Diaz et al. [21]. For this reason, we analyzed it separately (see Tables A1 and A2).

### 2.3.5. Visions of Local Communities Regarding to the Future of the CCLC

To explore how farmers and their families perceive the future of the rural landscape in the CCLC, we asked how do you imagine the future of the rural landscape? Responses were classified according to three categories: disappearance of rural areas, displacement, and uncertainty due to change. Additionally, we asked farmers to express motivation underpinning their responses, which were classified as both direct and indirect drivers of global change [41,42], including sociopolitical change and land-use change, as well as economic, cultural, and climate change (Figure 3). Two direct drivers were mainly recognized as change promoters in the region, i.e., land-use change and climate change. Additionally, we recognized visions associated with three indirect drivers: economy, political, and culture [43]. The economy was defined as per capita income and the taxes and subsidies provided by the government; the political reasons were defined as the mechanisms for the development of the rural sector; the culture was determined as values, beliefs, and norms that a group of people share.

## 3. Results

### 3.1. Farmers and Farms' Characteristics

Farmers interviewed were mainly from Risaralda (15 farmers), Valle del Cauca (5), Caldas (3), Quindío (2), and Antioquia (1). Only one farmer did not express its place of origin. The interviewees were made up of farm owners (48%), owner-managers (26%), managers (19%), and farm workers (7%). The age of the interviewees ranged from 26 to 85 years old, and the time spent in the region ranged from 3 to 69 years.

We found that 33.3% of farmers were female. It was also found that 44.4% of them were between 60 and 85 years of age (elderly). The educational level was heterogeneous, with 33% of farmers with no studies, 22% with elementary school, 19% with high school, 11% with a university degree (15% of responses were not registered). We found that the most common family type was the extended family (i.e., more family members live in the household, such as grandparents, aunts, uncles, cousins, etc.), followed by the nuclear family (parents and children). Regarding the farms' characteristics, we found that 66.7% of the farms showed changes in land use between 1997 and 2014, the most dominant being a

land transition from coffee to heterogeneous agriculture practices. We also found that some farms persisted despite being located in urban cover areas. Finally, 55.6% of the farms were located far from the urban area (Table 1).

**Table 1.** Farmers and farms' characteristics in the CCLC.

| | Variable | Category | Range | *n* | Average | Used in MCA |
|---|---|---|---|---|---|---|
| Farmers | Gender | Female | | 9 | 33% | √ |
| | | Male | | 18 | 67% | |
| | Age (years) | Adult | 26–59 | 15 | 56% | √ |
| | | Elderly | 60–85 | 12 | 44% | |
| | Relationship with the farm | Managers | | 5 | 19% | |
| | | Owner-managers | | 7 | 26% | |
| | | Owners | | 13 | 48% | |
| | | Workers | | 2 | 7% | |
| | Educational level | No data | | 4 | 15% | |
| | | No study | | 9 | 33% | |
| | | Primary school | | 6 | 22% | |
| | | High school | | 3 | 11% | |
| | | Technical | | 1 | 4% | |
| | | Technology | | 1 | 4% | |
| | | University degree | | 3 | 11% | |
| | Family type | Childless couples | | 3 | 11% | |
| | | Extended | | 12 | 44% | |
| | | NA | | 2 | 7% | |
| | | Nuclear | | 10 | 37% | |
| Farms | Altitude (m.a.s.l.) | Low | 1221–1572 | 16 | 59% | |
| | | High | 1573–1922 | 11 | 41% | |
| | Area (ha) | <14 ha | 0.5–14 | 23 | 85% | √ |
| | | >14 ha | 14–28.8 | 4 | 15% | |
| | Type of crops * | Traditional | | 20 | 74% | √ |
| | | Innovative | | 7 | 26% | |
| | Type of crops generating income ** | Monoculture | | 7 | 26% | |
| | | Subsidiary | | 20 | 74% | |
| | Time on the farm (years) | >36 | 36–69 | 14 | 52% | |
| | | <36 | 26–36 | 13 | 48% | |
| | Hiring personnel | No | | 11 | 41% | |
| | | Yes | | 16 | 59% | |
| | Destination of the production | Sale and self-consumption | | 17 | 63% | √ |
| | | Sale | | 10 | 37% | |
| | Farming style | Peasant | | 18 | 67% | |
| | | Semi-industrial | | 9 | 33% | |
| | Land-cover change in 1997–2014 | Yes | | 18 | 67% | |
| | | No | | 9 | 33% | |
| | Distance | Near | 18,957–38,530 | 12 | 44% | √ |
| | | Far | 38,531–58,102 | 15 | 56% | |

* Type of crops: Traditional, are defined as those crops that have always been cultivated in the area of the farm (coffee, banana and citrus); Innovative, refers to crops that have not been traditional in the area, are new to the area of study (tropical flowers, succulents, vegetables). ** Type of crops generate income: It is related to the type and number of crops that provide the economic income for the farm. Monoculture: one crop; Subsidiary: Several crops contribute to income.

Two farming styles were identified, 66.7% with arrangements tending towards peasant and 33.3% towards semi-industrial farms (Table 1). The peasant style was characterized by farms with an area of less than 14 ha, with traditional crops, crop association, no hired personnel, and production destined for sale and self-consumption. In addition, in the peasant style, the person in charge of the farm's activities and administration was the owner or an administrator who had been on the farm for more than 37 years. On the other hand, farms with a semi-industrial style were represented by farms with more than 14 ha,

dominated by novel monocultures, with hired personnel for field work and the production was destined for sale. Additionally, we found that the person in charge of the farm was an administrator or hired worker who had been with the farm for less than 36 years.

The MCA differentiated significant associations between farmers' and farms' characteristics. Dimension 1 identified the relationship between the variables farm with area greater than 14 ha and destination of the production for sale, while in dimension 2, the variables that contributed the most were crop type, monoculture, and distance near and far; in dimension 3, they were female and male genders (Figures 4 and A2–A4). The first three dimensions explained 76.1% of the variance. We found an associated statistical significance in dimension 1 (36.7% of the variance) and in dimension 2 (26.3%). On the *X*-axis (dimension 1), we found a good separation of farms according to area and production destination. On the *Y*-axis (dimension 2), the farms were distributed in relation to type of crop and distance (Figures 4 and A1). Farmers older than 60 years old were mainly female and their production was for sale and self-consumption.

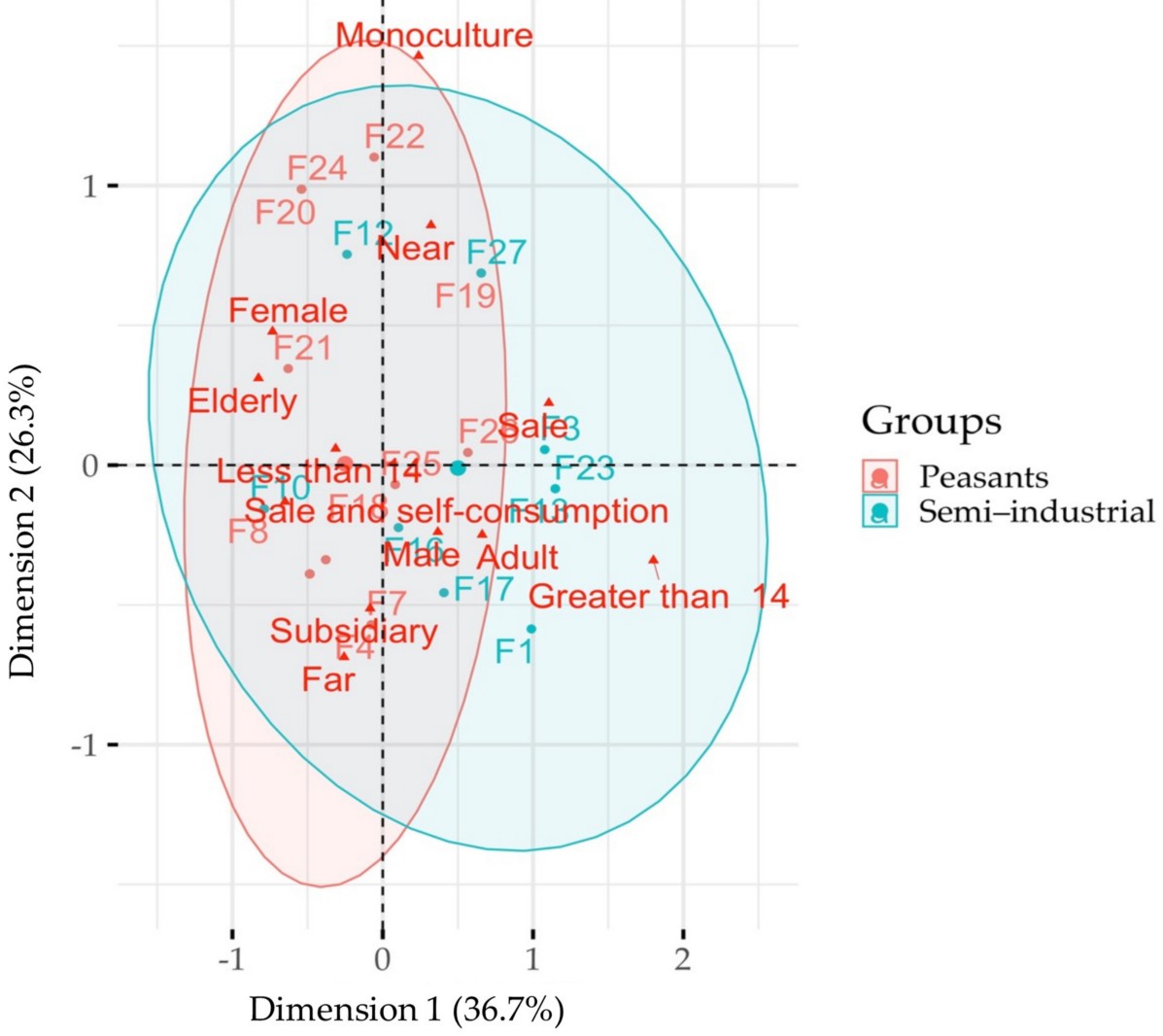

**Figure 4.** Multiple correspondence analysis of farmers' and farms' characteristics.

### 3.2. Diversity of Emotions Associated with Farmlands

Results showed that farmers identified multiples emotions associated with living on the farmlands of the CCLC. Examples of these emotions included "the farm is a lot of peace, silence and tranquility" (tranquility); "The farm makes my soul happy" (happiness); "I don't know. I feel nostalgia when I work in the fields because I remember my father, I

imagine him working there" (rootedness); "The farm generates security" (safety); "The farm is wonderful" (awe); "The farm is life, I breathe pure and clean air" (vitality); "The farm is freedom" (freedom); and "Through the work on the farm I think and begin to philosophize" (interest). According to the classification of the emotions used, we found that tranquility (69%), happiness (31%), rootedness (27%), and safety (23%) were the most common emotions or feelings associated with farm landscapes, followed by awe and vitality (15%), freedom (12%), and interest (12%) (Figure 5).

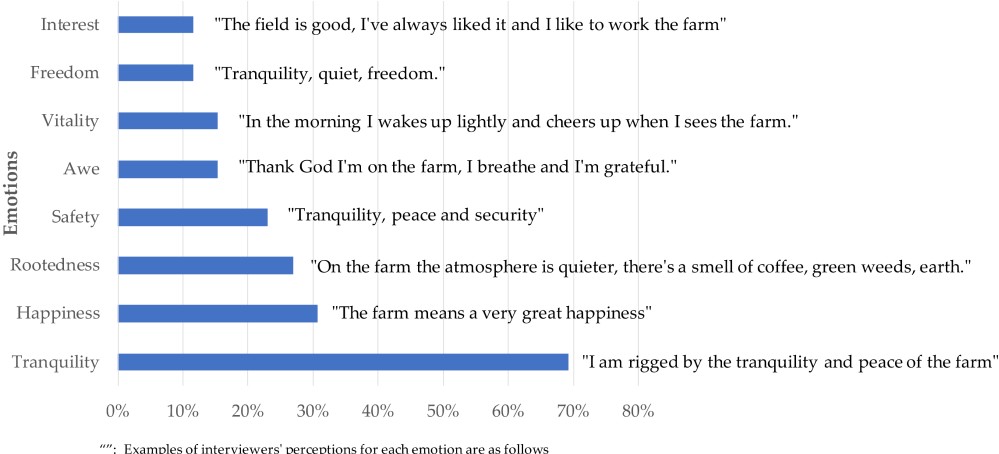

"": Examples of interviewers' perceptions for each emotion are as follows

**Figure 5.** Recognition of emotions generated by living on the farm.

We found that gender and age were significantly related to family rootedness ($p < 0.05$ for gender and $p < 0.10$ for age). Rootedness is understood as the affective bond they have in accordance with the identity to the farm. Additionally, according to the farming style, we found a significant relationship between the contribution of the farm to human safety ($p < 0.05$), tranquility ($p < 0.1$) and happiness, admiration and vitality ($p < 0.15$). Finally, we also found a correlation between the farm distance to urban areas and the perceptions regarding farm rootedness and safety ($p < 0.15$ for both emotions) (Table 2).

**Table 2.** Variables that influence the different types of senses on the farm.

| Variables | Tranquility | Happiness | Rootedness | Safety | Awe | Vitality | Freedom | Interest |
|---|---|---|---|---|---|---|---|---|
| Farming style | ** | * | *** | | | * | * | |
| H of Kruskal–Wallis | 2.889 | 2.138 | 8.357 | 1.486 | 0.000 | 2.261 | 2.261 | 1.625 |
| Degree of freedom | 1 | 1 | 1 | 1 | 1 | 1 | 1 | 1 |
| Two-sided *p*-value | 0.089 | 0.144 | 0.004 | 0.223 | 1.000 | 0.133 | 0.133 | 0.202 |
| Distance | | | | * | * | | | |
| H of Kruskal–Wallis | 0.000 | 0.214 | 2.321 | 2.684 | 0.650 | 0.057 | 0.057 | 0.650 |
| Degree of freedom | 1 | 1 | 1 | 1 | 1 | 1 | 1 | 1 |
| Two-sided *p*-value | 1.000 | 0.644 | 0.128 | 0.101 | 0.420 | 0.812 | 0.812 | 0.420 |
| Gender | | | | *** | | | | |
| H of Kruskal–Wallis | 0.722 | 1.368 | 0.929 | 4.550 | 1.625 | 0.565 | 0.141 | 1.625 |
| Degree of freedom | 1 | 1 | 1 | 1 | 1 | 1 | 1 | 1 |
| Two-sided *p*-value | 0.395 | 0.242 | 0.335 | 0.033 | 0.202 | 0.452 | 0.707 | 0.202 |
| Age | | | | ** | | | | |
| H of Kruskal–Wallis | 0.000 | 1.445 | 0.371 | 3.352 | 0.163 | 1.710 | 0.057 | 0.163 |
| Degree of freedom | 1 | 1 | 1 | 1 | 1 | 1 | 1 | 1 |
| Two-sided *p*-value | 1.000 | 0.229 | 0.542 | 0.067 | 0.687 | 0.191 | 0.812 | 0.687 |

Signification of codes: 0.05, '***'; 0.1, '**'; 0.15, '*'.

### 3.3. Sense of Place of Local Communities in the CCLC

Regarding the farmers' motivation to remain on these farm landscapes in the near future, we found that 85% of the farmers expressed a positive motivation to remain in

the CCLC, while 11% of farmers responded negatively, and 4% felt uncertainty. The most frequent motivations for remaining in this region were associated with the recognition of farms as their place (85%), followed by pleasure and well-being of living there (37%), a collective recognition of the countryside as a home (identity) (33%), tranquility, air quality (clean and no noise), and the freedom of being in open spaces (22%). To a lesser extent, we also found labor (19%), fullness (15%), and farming security (11%) to be important motivations to remain in the region (Figure 6).

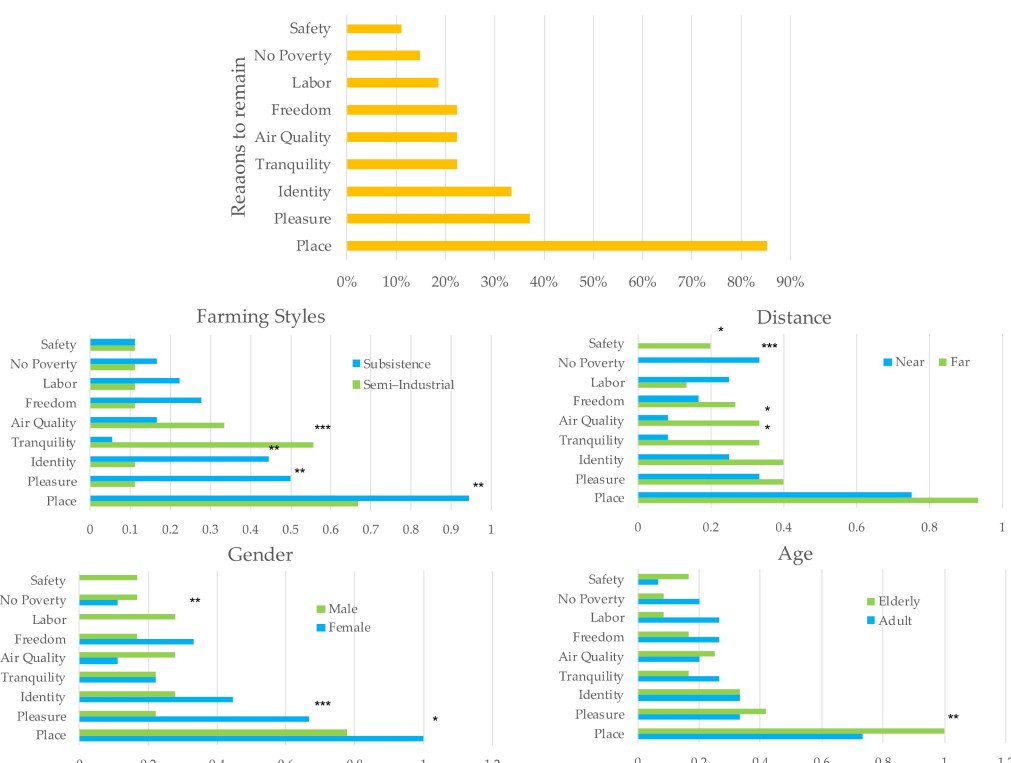

**Figure 6.** Farmer's motivations to remaining in the CCLC. Signification of codes: 0.05, '***'; 0.1, '**'; 0.15, '*'.

Regarding gender, we found that men were strongly connected to farm tasks ($p < 0.1$), while women mainly valued being on the farm the most ($p < 0.05$) and the recognition of the farms as a home ($p < 0.15$). Regarding age, we found that older adults (over 60 years old) were more willing to remain on the farm due to the recognition of the farm as a place to live ($p < 0.05$) (Figure 6).

We also found that farms with a semi-intensive farming style valued tranquility more than farms with a peasant style ($p < 0.05$). However, peasant farms recognized farms as dwelling, providing pleasure and identity ($p < 0.1$) as motivations to remain. Regarding the distance to urban areas, we observed that the farms closer to the urbanized areas showed motivations to remain associated with fullness ($p < 0.05$). On the contrary, farms located farther were more associated with benefits linked to tranquility, air quality, and security ($p < 0.15$) (Figure 6).

### 3.4. Nature's Contributions to People in the CCLC

Of the eighteen NCPs, farmers identified seven NCPs associated with the farm landscapes of the CCLC (Figure 7). We found that nonmaterial NCP were the most commonly associated with farmlands, including physical and psychological experiences (NCP16, 85%), maintenance of options (NCP18, 74%), and supportive identities (NCP17 56%). We also found regulating NCP such as habitat creation and maintenance (NCP1, 52%) and air quality regulation (15%) to be important contributions in this region.

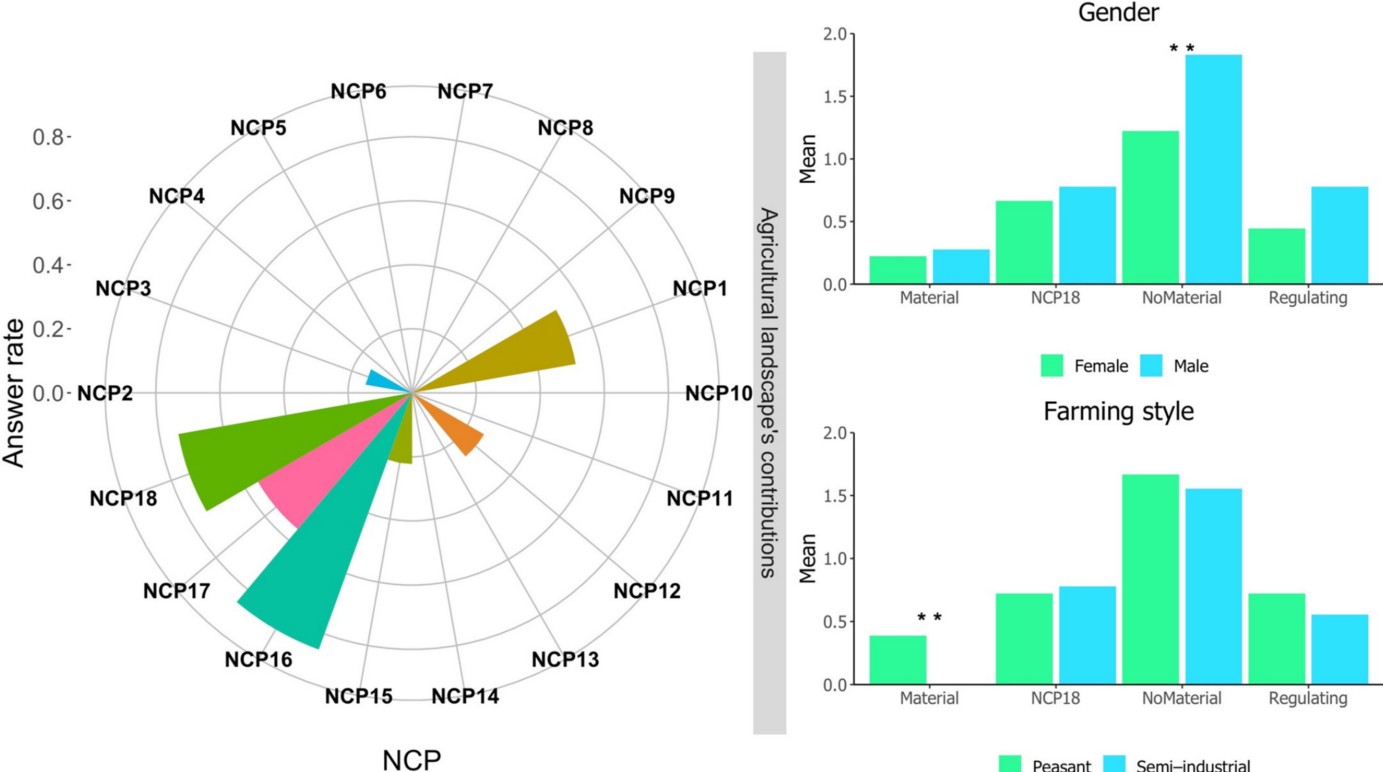

**Figure 7.** Farmer's perception of NCP provided by farms in the CCLC. NCP1, habitat creation and maintenance; NCP2, pollination and dispersal of seeds and other propagules; NCP3, regulation of air quality; NCP4, regulation of climate; NCP5, regulation of ocean acidification; NCP6 regulation of freshwater quantity, location, and timing; NCP7, regulation of freshwater and coastal water quality; NCP8, formation, protection, and decontamination of soils and sediments; NCP9, regulation of hazards and extreme events; NCP10, regulation of detrimental organisms and biological processes; NCP11, energy; NCP12, food and feed; NCP13, materials, companionship, and labor; NCP14, medicinal, biochemical, and genetic resources; NCP15, learning and inspiration; NCP16, physical and psychological experiences; NCP17, supporting identities; NCP18, maintenance of options. Signification of codes: 0.1, '**'.

We found significant differences in the mean response for nonmaterial and material NCP across gender and farming style. Male identified more nonmaterial NCP than women ($p < 0.1$) (Figure 7). Male recognized farms as spaces where identities are supported, a source of satisfaction and experiences, family rootedness, and agricultural traditions. Among the stories recorded, we found examples such as "Every night there is a longing for the work of the other day" (rural man, 71 years old); "All my life I have lived in the countryside, I have always liked it. And in the area, everything is very quiet, it is safe" (rural man, 72 years old); "The farm gives me tranquility and brings back memories of my childhood, of my tradition. And it is also safe" (rural man, 26 years old).

Regarding farming styles, we found significant differences in relation to the material NCP ($p < 0.05$). In this sense, farms with peasant farming styles identified the importance to secure food for families. An example of stories reflecting this is: "On the farm there is always food within reach and there is no money involved" (rural woman, 37 years old); in the peasant style the production of the farm is destined both for sale and for self-consumption; on the contrary, farms with a semi-intensive style orient all their production for sale and do not recognize these material contributions of the agricultural landscape to the well-being of the rural family (Figure 7).

### 3.5. Visions of Local Communities Regarding the Future of the CCLC

Diverse visions were found associated with future changes in land use, including "the growth of the city" and "destruction of the natural environment for urban expansion", while the climate change was mostly recognized with visions such as "the change that has occurred in the rural sector has been mainly due to climate change" and "changes in the climate are quite perceived, the rainy and sunny seasons are more intense". We found visions related to "rural work is very hard and poorly paid", "rural people want to go to the city in search of better opportunities", "agricultural production is not profitable" and "the government will not let agricultural production end" are reasons included in the economy category. The political visions found were mainly related to "farmer is unprotected, has no social security", "the government does not support the field for lack of regulation and protection" and "the promotion of sustainable tourism with the people of the area". Moreover, the visions linked to culture were related to arguments such as "young people do not want to continue with the farm and work it" and "there is no one to work the land".

Of all visions found, 41.7% of farmers considered that the rural areas will disappear in the near future, while 33.3% of them expressed uncertainty and 25% believed that displacement to another site was the most likely option. Forty-seven reasons were collected supporting these visions of the future of the CCLC, mostly justified by arguments related to changes in land uses (27.7% of farmers), followed by economic arguments (23.4%), and sociopolitical and cultural arguments (21.3%). Farmers recognized climate change as a lesser force for future changes in rural areas, with 6.4% (Figure 8).

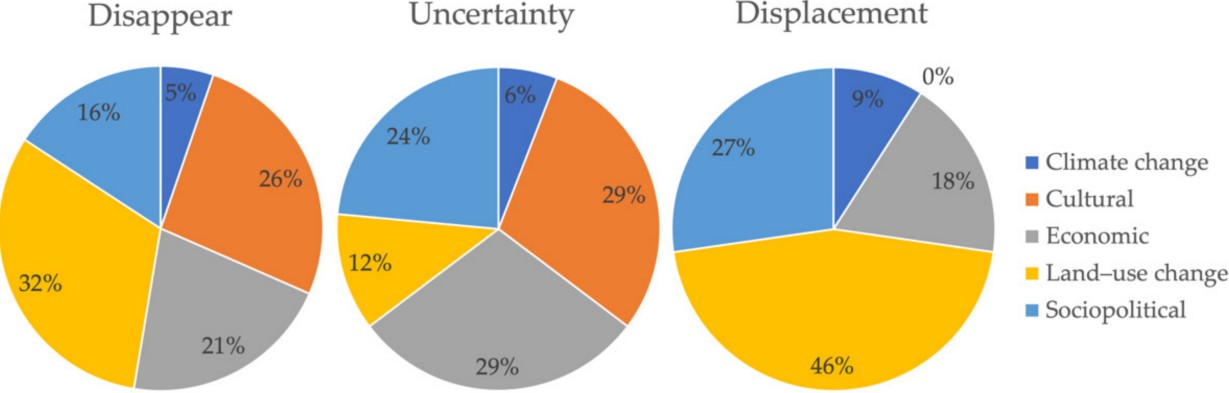

**Figure 8.** Farmers' perspectives and supporting arguments regarding the future of CCLC.

## 4. Discussion

### 4.1. Farmers and Farms' Characteristics in the CCLC

Our results identified nonmaterial NCP (e.g., as physical and psychological experiences maintenance of options and supportive identities) to be the most important contributions shaping the sense of place of farmers in the CCLC. This is consistent with several studies that have shown the long history of how rural families have developed cultural roots and have coevolved with farming landscapes in multiple intangible ways and forms [9,13]. Additionally, we found that farms' characteristics (e.g., farming styles, distance to cities, and gender) may play an important role in articulating farmers' meanings and attachment to these farmlands. In this sense, different farming styles appeared to be associated with the particular meanings and perceptions that farmers hold to the territory. This finding is consistent with several other studies where inhabited places reflected people's values, histories, material, and symbolic practices [16,17,32], thus indicating the importance of farming practices in shaping different levels of human connection to nature and in forming land stewardship [44–47]. Specifically, we found that two farming styles, peasant and semi-industrial, are shown to be influencing the farmers' perception toward particular NCP and emotions associated with farmlands (Figure 7, Table 2), and with motivations to remain in the CCLC (Figures 6 and 7). These results are consistent with findings in the

study Heterogeneity reconsidered [11], that showed the importance of comanagement of territory with communities for promoting land transitions that preserve and shape the sense of place within the land.

We found that gender played an important role related to the emergence of "pluriactivity" in farmlands, which in the theories of the new rurality, stands out as the incursion of women to generate income in especially nonagricultural activities. This is a relevant result because it changes the configuration of the sense of place, incorporates into future analyses the perspective of gender equity and the participation of different social actors in development processes and projects. Then, the examination of the role of NCP in the configuration of the farmers' sense of place in the CCLC allowed an inquiry about the new family configuration with increasing participation of women (33%), which assigns new functions to rural spaces in the ways of perceiving material, nonmaterial and regulatory NCP (Figures 4, 6 and 7). A gender-inclusive analysis showed that men and women often value NCP in different ways and may possess diverse knowledge, with implications for the value of places for management priorities [48] and the formulation and implementation of sustainable and equitable policies and interventions [49].

### 4.2. Sense of Place in the CCLC

Sense of place is defined as the meanings and attachments that people possess in a territory [12,16,17] (Figure 9). Our findings were able to identify specific NCP underpinning the diversity meanings and the attachment of Pereira's farmers to the farmlands in the CCLC.

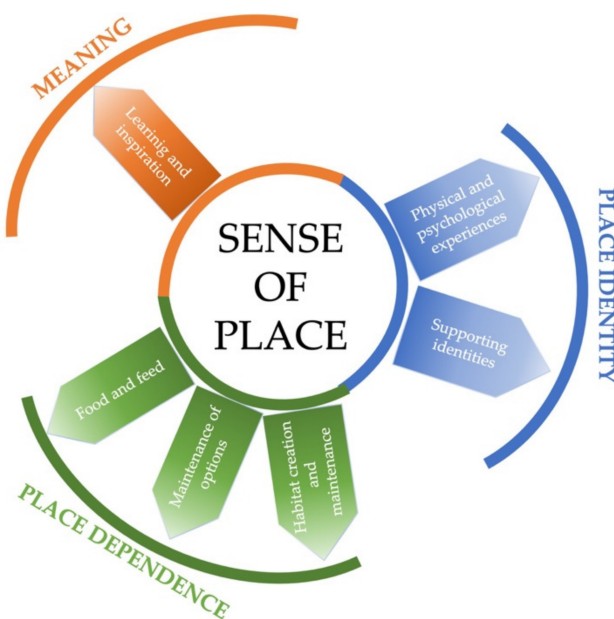

**Figure 9.** Characterization of the sense of place in the CCLC (adapted from Masterson et al. [16]).

Firstly, the diversity of meanings found were mainly interpreted through the diversity of emotions towards farmlands and the opportunities associated with learning and inspiration (NCP 15). Examples of these emotions included tranquility, happiness, freedom, and interest, and can be interpreted as reflections of farmers' experiences of living in the CCLC. This result is aligned with findings of Rajala and Sorice [12] that showed how landowners' emotions can contribute directly to farmers' emotional health.

Secondly, as defined by [50], attachment to a place is developed through daily and sustained interactions, as well as a strong motivation to maintain the relationship with a place over time. Our findings identified a relationship between farmers and place in the CCLC (i.e., interpreted the place identity and dependence). Place identity defines an individual's personal identity with the physical environment [16,17]. Our study captured

the reasonings for remaining in the CCLC such as "I am a peasant and very proud of my roots", which reflects on how a person's identity is linked to a place and depends on specific farmland contributions, such as supporting identity (NCP17) and psychological experiences (NCP16) associated with living on a farm. Here, we argue that this qualitative information must be used for the understanding of identity and attachment along agricultural landscapes [21]. These findings also support recent insights that have shown how the landscapes of the CCLC are intrinsically connected with cultural assets and meanings ascribed to farmlands [19,21,32].

The place dependency to farmlands conveys an instrumental connection between people and place, conceived and measured as the capacity of an environment to facilitate the achievement of goals and satisfy important needs [16]. Our results found that the most important material nature's contribution to people identified in farms of the CCLC was food production (NCP 12), which is crucial to sustain livelihoods of local communities. In addition, one of the strongest reasons given by farmers to remain on the farmlands of the CCLC was the place where they inhabit themselves, which can be interpreted as a way to recognize the capacity of this region to provide security and support tranquility of livelihoods (Figure 6). Another example of place dependency bonds to the land was revealed to be communities' perceptions of farms as a space that maintains the options for a good quality of life (NCP 18) and as a place where they have been able and can continue to develop their livelihoods and persist. This may reflect how place dependence enhances place identity and in turn influences people's responsible behavior [46,50]. Here, we argue that this finding can be interpreted as evidence that farmers do not perceive themselves as separated from their farms in the Pereira CCLC.

Finally, results obtained in this study must be interpreted in the context of some limitations. First, one limitation had to do with the impossibility of sampling a larger number of farms and farmers due to the lack of financial resources and the need for additional fieldwork research assistants. Second, another limitation had to do with the difficulty in building trust with farmers, which influenced our ability to run more extensive interviews and obtain more precise information regarding the institutional aspects of farmland governance in the CCLC. Finally, the lack of security in the Pereira region greatly hindered sampling efforts in the study due to the local communities' distrust of visitors or foreigners.

## 5. Conclusions

UNESCO recognized in 2011 the CCLC as a world heritage site, which influenced Colombia laws and management plans for its preservation and care. This study provides empirical evidence of the important role that nature's contributions to people play in shaping the sense of place and land heritage in the CCLC. The diverse farms studied in the CCLC showed how the heterogeneity of farming styles are key for preserving biocultural diversity of this region, which demonstrates the strong relationship between sense of place and human behavior and provides evidence that affective attachment to lands can shape behavior towards nature protection. However, progressing on this direction requires time to build trust with farmers and financial and human resources to create collective planning strategies. Future work must address the need for robust and transparent deliberative and negotiation mechanisms that are inclusive of all relevant stakeholders (i.e., their perceptions and cultural differences), aim to address unequal power, and recognize and strengthen communities' governance within the CCLC.

**Author Contributions:** Conceptualization, B.E.M.-L., A.J.C and A.F.-M.; methodology, B.E.M.-L. and A.J.C.; formal analysis, B.E.M.-L.; investigation, B.E.M.-L. and A.F.-M.; writing—original draft preparation, B.E.M.-L.; writing—review and editing, A.J.C. and A.F.-M.; visualization, B.E.M.-L.; supervision, A.F.-M. and A.J.C. All authors have read and agreed to the published version of the manuscript.

**Funding:** The doctoral thesis was supported by the internal funds of the Vice-Rectory of Research of the Technological University of Pereira, Colombia (E2-18-2). The first author was funded by

a scholarship of COLCIENCIA-COLFUTURO (647), Colombia and a mobility grant funded by the AUIP.

**Institutional Review Board Statement:** Not applicable.

**Informed Consent Statement:** Informed consent was obtained from all subjects involved in the study. The authors were authorized by all interviewees based on Law 1581 of 2012, article 9 and 12, Regulatory Decree 1377 of 2013 article 10 of Colombia.

**Data Availability Statement:** Not applicable.

**Acknowledgments:** We thank all the farmers who kindly allowed us to enter their farms and who generously responded to our semistructured interview. This publication is part of the doctoral thesis in environmental sciences of the first author in the Graduate Programme of the Faculty of Environmental Sciences of the Technological University of Pereira. The doctoral thesis was supported by the internal convocation of the Vice-Rectory of Research of the Technological University of Pereira, Colombia (E2-18-2). The first author had a forgivable loan scholarship from COLCIENCIA-COLFUTURO for her doctoral studies (647), Colombia. This publication was funded by UAL. Finally, the authors appreciate the participation and support provided by the research group Socio-Ecological Research Lab at the University of Almeria and the Management in Tropical Andean Agroecosystems (GATA—Spanish acronym).

**Conflicts of Interest:** The authors declare no conflict of interest.

## Appendix A

**Table A1.** Definition and summary of variables.

| Objectives | Variables | Variable Definition. Brief Explanation |
|---|---|---|
| Farmers and farms' characteristics | Gender | Gender of the interviewed farmer |
| | Age (Years) | Age of the interviewed farmer |
| | Farming style | Comprise ways of organizing and reorganizing the internal and external requirements of the farms and are firmly rooted in a stock of cultural knowledge |
| | Land-use change | Whether or not there was a change in land cover around the farms between 1997 and 2014 |
| | Distance | Distance variable as near and far from the most central collection center in the city |
| Human Emotions | Tranquility | Quality or state of being tranquil; calmness; peacefulness; quiet; serenity; free from or unaffected by disturbing emotions; unagitated; serene; placid |
| | Happiness | State of pleasant spiritual and physical satisfaction |
| | Rootedness | An affection, a virtue, a use or a habit: to become very firm; to establish oneself permanently in a place, binding oneself to people and things |
| | Safety | Quality of a site that provides security, certainty, confidence |
| | Awe | To see, contemplate or consider with special esteem or pleasure something that calls our attention because of qualities judged as extraordinary. |
| | Vitality | 1. f. Quality of having life; 2. f. activity or efficiency of the vital faculties (quality of life) |
| | Freedom | The natural ability of people to act in one way or another, and not to act, so they are responsible for their actions |
| | Interest | Inclination or attraction felt towards an object or activity they like; activity that is done habitually and for pleasure in leisure time |

**Table A1.** *Cont.*

| Objectives | Variables | Variable Definition. Brief Explanation |
|---|---|---|
| Reasons to remain | Place | Farmers define it as the space in which their place is; the farm is more than that space for agricultural production; it is the personal relationship with the territory, where production takes place, where the family lives and is formed |
| | Pleasure | Pleasure is related to the feeling of well-being generated by staying on the farm and not being in the city |
| | Identity | Farmers are defined in relation to the farm, the rural life, working in the field and being a farmer; it is related to the roots and tradition |
| | Tranquility | Quality or state of being tranquil; calmness; peacefulness; quiet; serenity. Free from or unaffected by disturbing emotions; unagitated; serene; placid. |
| | Air quality | They express that the air on the farm is clean |
| | Freedom | The natural ability of people to act in one way or another, and not to act, so they are responsible for their actions. Farmers express freedom on the farm as open space, open doors and windows; the possibility of going from one place to another without restrictions in the space itself |
| | Labor | Related to always having something to do, being busy, and feeling useful |
| | No poverty | They express that there is never a lack of food on the farm no matter how difficult the situation |
| | Safety | Quality of a site that provides security, certainty, confidence |
| | Support community | Strength in the relationship with the community; the neighborhood that exists; the support and care provided to each other |
| Nature's contributions to people (NCP) | Habitat creation and maintenance | " . . . conditions necessary or favorable for living beings of direct or indirect importance to humans" |
| | Regulation of air quality | Perception " . . . Filtration, fixation, degradation or storage of pollutants that directly affect human health or infrastructure" |
| | Food and feed | "Production of food from wild managed, or domesticated organisms" |
| | Learning and inspiration | "Provision, by landscapes, seascapes, habitats or organisms, of opportunities for the development of the capabilities that allow humans to prosper through education, acquisition of knowledge and development of skills for well-being, information, and inspiration for art and technological design" |
| | Physical and psychological experiences | "Provision, by landscapes, seascapes, habitats or organisms, of opportunities for physically and psychologically beneficial activities, healing, relaxation, recreation, leisure, tourism and aesthetic enjoyment based on the close contact with nature" |
| | Supporting identities | Landscapes, seascapes, habitats or organisms being the basis for religious, spiritual, and social-cohesion experiences; source of satisfaction derived from knowing that a particular landscape, seascape, habitat or species exists |
| | Maintenance of options | Capacity of ecosystems, habitats, species or genotypes to keep options open in order to support a good quality of life |

## Appendix B

*Methodological Approach*

We investigated the role of the diversity of NCP in shaping the sense of place in the CCLC. We followed the approach of Masterson et al. (2017) where place attachment and place meanings are described as key concept to understand the motivation for stewardship and actions to care for the environment and use the resources.

In this sense, (i) the farmers' emotions associated with farm landscapes are connected with place meanings; (ii) the reasons of local communities for remaining on the farm, approaches to place attachment and (iii) the diversity of nature's contributions to people provided by farm landscapes are used to try to explain both meanings and attachment to place (Table A2).

**Table A2.** Questions, quotes and codes.

| Objective | Questions | Examples of Responses | Interpretation Source | Code |
|---|---|---|---|---|
| Emotion | What do you like most about living here on your farm? What feelings or sensations do you have about the farm? What do you think about the place where you live? | "I don't know. I feel nostalgic when I work in the fields because I remember my father, I imagine him working over there" "The farm generates a feeling of tranquility, happiness to the soul and security" "I am happy to see this beautiful place. The morning rises lightly and perks up when you see the farm" | Responses were interpreted and classified as different emotions that emerged from the dialogue. | Tranquility Happiness Rootedness Safety Awe Vitality Freedom Interest |
| Reasons to remain | Do you like living on the farm? Why do you remain on the farm? | "To live in the field because I don't like to live in the city, because in the city there are bad influences. I would not like to live in the city because there is nothing to do there, I would only interfere with the family" "Life in the city is very busy. Here in the field everything is peaceful, I only worry about having my plants well and if I want a banana or a mango, the land itself gives them to me" | Responses were interpreted and classified as different reasons to remain on the farm that emerged from the dialogue. | Place Pleasure Identity Tranquility Quality air Freedom Labor No poverty Security Support community |
| Nature's contributions to people | What do you like most about living on your farm? What does the farm offer you? | " … I always like to be working the field"; "It is easier to educate children in the field, neighbors take care of all children" (NCP code: Learning and inspiration. NCP15); "It is a calm and healthy life"; "The farm gives me tranquility" (NCP code: Physical and psychological experiences. NCP 16); "I consider myself a peasant of -pura cepa- and very proud, it's my rootedness "; "I like the farm because I was born there, it is a matter of tradition" (NCP code: Supporting identities. NCP 17) "You find everything on the farm. I like everything on the farm, living on it because you live in peace" (NCP code: Maintenance of options. NCP 18) | Responses were interpreted and classified as different NCP that emerged from the dialogue. We use the framework developed by Díaz et al. 2018 | NCP1, habitat creation and maintenance; NCP3, regulation of air quality; NCP12, food and feed; NCP15, learning and inspiration; NCP16, physical and psychological experiences; NCP17, supporting identities; NCP18, maintenance of options |

**Appendix C**

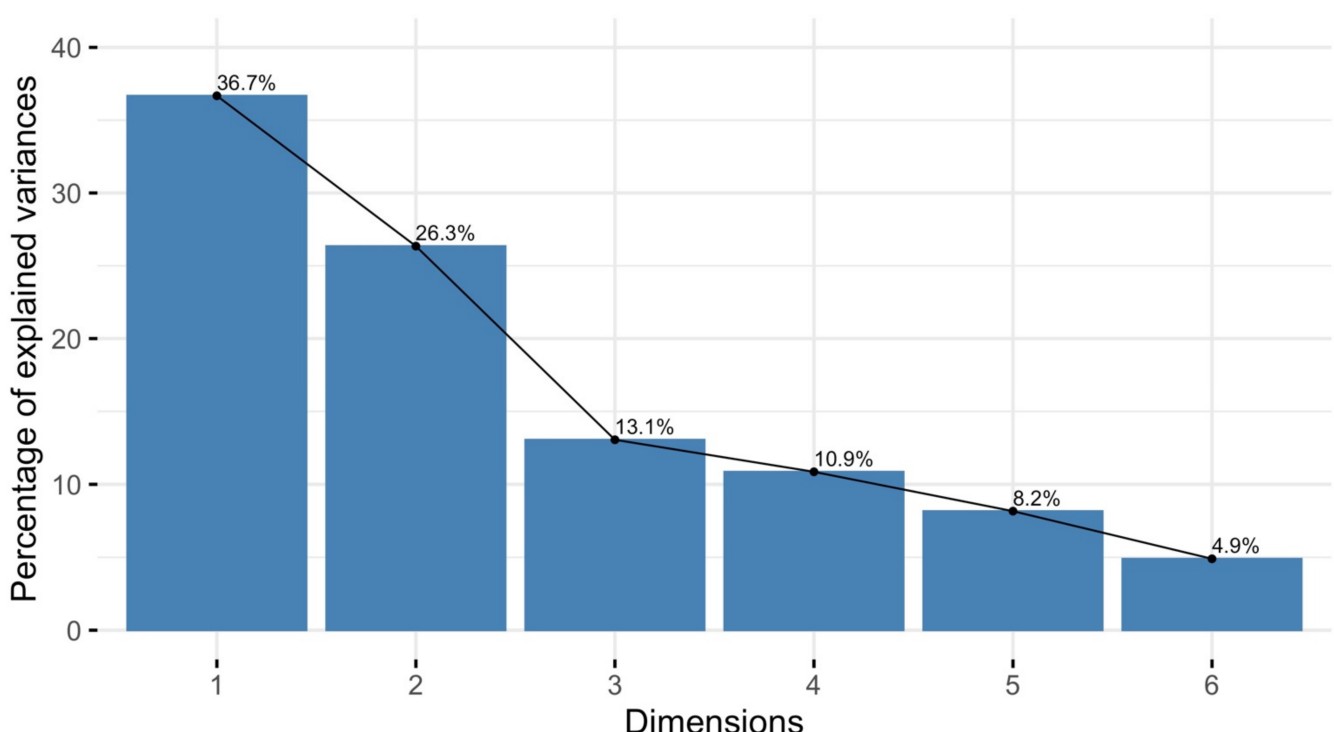

**Figure A1.** ACM's dimensions and percentage of explained variances.

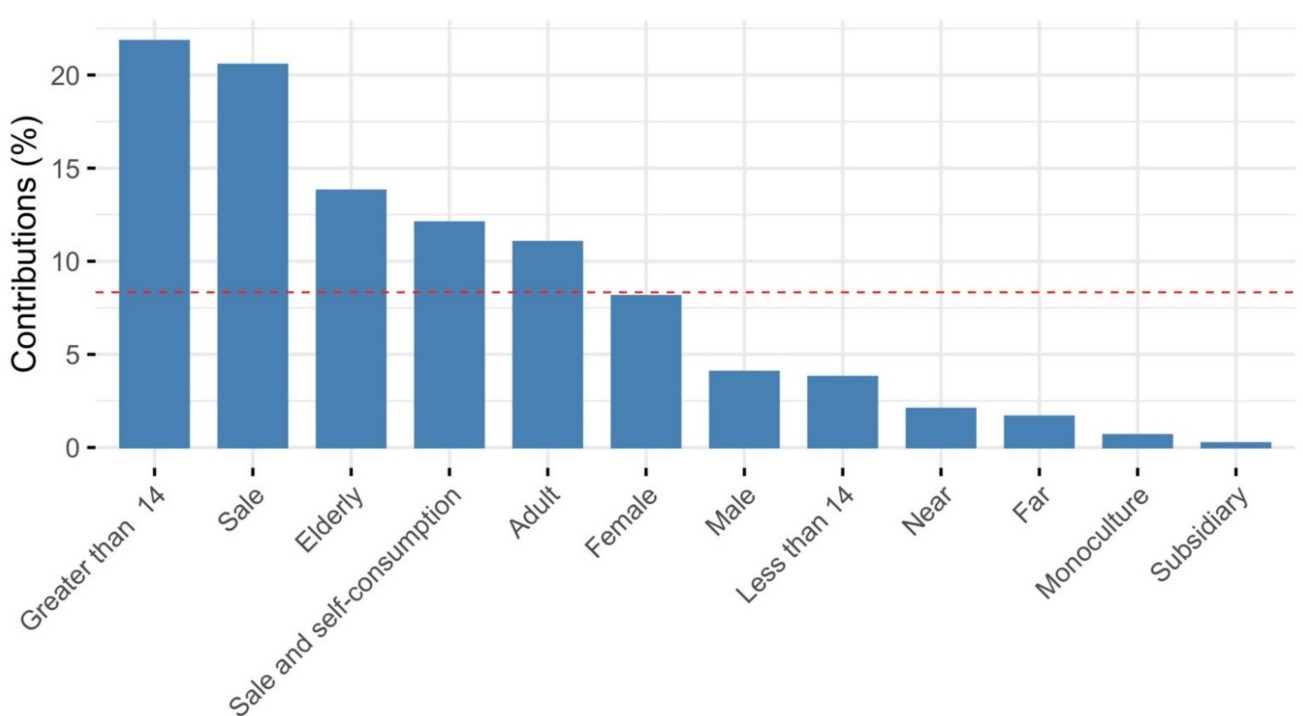

**Figure A2.** Contribution of variables to dimension 1.

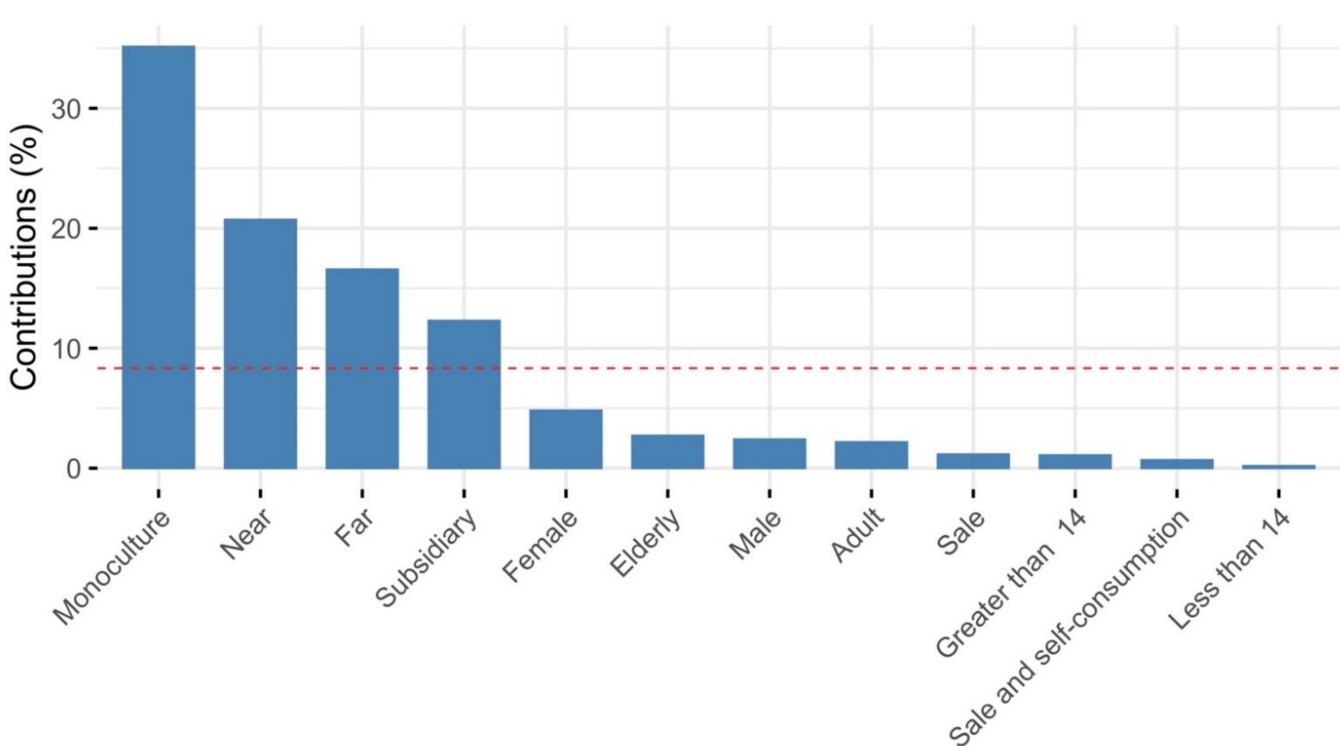

**Figure A3.** Contribution of variables to dimension 2.

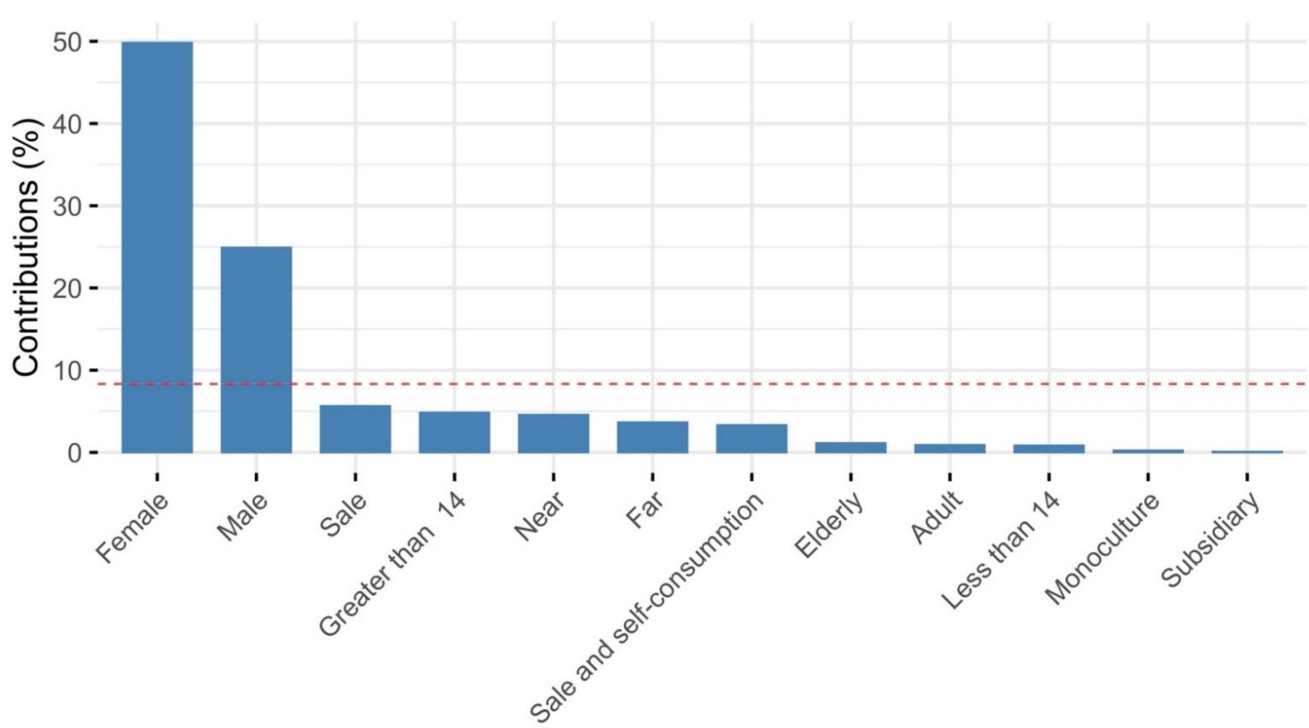

**Figure A4.** Contribution of variables to dimension 3.

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
