# Peer review of "Nature’s Contributions to People Shape Sense of Place in the Coffee Cultural Landscape of Colombia"

_agriculture, doi:10.3390/agriculture12040457_

Round 1
Reviewer 1 Report
The authors did an excellent job in the adjustments process. I do not have any concerns.
Author Response
We really appreciate your positive comments

Reviewer 2 Report
I found the paper very interesting and the issue addressed is certainly very important due to the global challenges rural world is facing. I am not sure that the term “nature” perfectly fits, as we are talking about a cultural landscape, not about a natural one. Probably the term ecosystem services should be introduced in place of NCP.
Some minor comments:
42-43 authors refers to transoformations involving specific land uses (gallery and/or riparian forest and bamboo forest) that clearly refers to the study site or more in general to the cultural landscape of coffee cultivation, but without explaining that they are talking of the study site. the sentence seems a bit impromptu as it is inserted in a generic paragraph. I suggest to move it after the paragraph 47-54.
124, 152,. please replace “masl” with “m a.s.l.”
- Please, add the scale bar to the map.
189-195. Please, move these lines to the results section.
- Authors should add few lines explaining how the results of their survey could be used at local level for preserving the CCLC and if there are local territorial planning, management plan of the UNESCO site, local policies, are taking care of the safeguarding of traditional agriculture and therefore of the local cultural landscape.
Author Response
March 17th, 2022
Dear Assistant Editor and Reviewers,
Please find attached a revised version of our manuscript agriculture-1610671 entitled “Nature’s contributions to people shape sense of place in the Coffee Cultural Landscape of Colombia” to be considered for publication in Agriculture Special Issue. Overall, this revised version has been substantially reworded and now addresses minor comments provided by R1, R2 and R3, which are summarized in the following points:
- We have reworded specific sections according to recommendations and minor suggestions provided by R2 and R3.
- We have substantially reworded the conclusions sections in order to clearly explain and connect between obtained results and discussing points
- Figure 2, 3 and 8 are edited
- An English native writer has edited this revised version
For a clearer understanding, we attach our responses letter reproduces in table 1 are all comments provided by R2 and in table 2 are all comments provided by R3 and our responses/justification to each of them. We truly want to thank to both editor and reviewers for providing such a positive insights and helpful comments to improve our manuscript.
Sincerely,
Beatriz Elena Murillo-López and Antonio J. Castro, on behalf of the authors

Reviewer 3 Report
The manuscript addresses an interesting topic concerning, inter alia, the sense of place of farmers in the Coffee Cultural Landscape of Colombia. The Authors applied a quali-quantitative methodological approach based on semi-structured interviews, in person observation, and informal conversation. The manuscript is quite well structured. The method is rooted in scientific basis. The manuscript shows pitfalls that need to be addressed by the Authors. Please see the detailed report below.
Detailed report
Line 28: the acronym CCLC has not been defined.
Line 123: “masl”: meters above sea level? Please consider defining the abbreviation.
Figure 3: the figure is quite difficult to read. I would suggest the Authors provide a new clearer Figure 3.
Line 263: “A Kruskal Wallis analysis”. I would suggest the Authors add more details and references concerning the “Kruskal Wallis analysis”.
Line 275: “In these categories NCP18 was not included”. It is unclear the reason(s) why NCP18 has not been included. Please consider adding details.
Table 1. It is unclear how the ranges (concerning age, altitude, area, etc.) were chosen by the Authors.
Line 398: “We found significant around the rol of gender and”. Please check.
Lines 398-402: “We found significant around the rol of gender and farming style in recognizing non-material and material NCP. Regarding gender, men identified more non-material NCP than women (p<0.1) (Figure 7), recognized farms as scenarios where identities are supported, a source of satisfaction and experiences, family rootedness and agricultural traditions”. The meaning of this paragraph is unclear. Please consider rewording.
Line 448-449: “Climate change the driver less represented as a force of future changes with 6.4% (Figure 8)”. Please consider rewording.
Line 483: “new family configuration with high participation of women (33%)”. It seems that 33% does not represent high participation but perhaps moderate participation.
Line 510: “Place identity define”. Please check.
Section “5. Conclusions”: this section lacks limitations of the study. I would suggest the Authors clearly state the limitations of the study and the relevance of the study to the international audience.
Please check Table A1.
Lines 583-587: “In this sense, i. the farmer’s emotions associated to farm landscapes are connected with place meanings; ii. Sense of place of local communities to remaining in the farm approaches to place attachment; and iii. Diversity of nature's contributions to people provided by farm landscapes tries to explain both meanings and attachment to place (Table B1)”. Please consider rewording.
Please check the caption of Figure C1.
Author Response
March 17th, 2022
Dear Assistant Editor and Reviewers,
Please find attached a revised version of our manuscript agriculture-1610671 entitled “Nature’s contributions to people shape sense of place in the Coffee Cultural Landscape of Colombia” to be considered for publication in Agriculture Special Issue. Overall, this revised version has been substantially reworded and now addresses minor comments provided by R1, R2 and R3, which are summarized in the following points:
- We have reworded specific sections according to recommendations and minor suggestions provided by R2 and R3.
- We have substantially reworded the conclusions sections in order to clearly explain and connect between obtained results and discussing points
- Figure 2, 3 and 8 are edited
- An English native writer has edited this revised version
For a clearer understanding, we attach our responses letter reproduces in table 1 are all comments provided by R2 and in table 2 are all comments provided by R3 and our responses/justification to each of them. We truly want to thank to both editor and reviewers for providing such a positive insights and helpful comments to improve our manuscript.
Sincerely,
Beatriz Elena Murillo-López and Antonio J. Castro, on behalf of the authors

This manuscript is a resubmission of an earlier submission. The following is a list of the peer review reports and author responses from that submission.
Round 1
Reviewer 1 Report
Title: Nature’s contributions to people shape sense of place in the Coffee Cultural Landscape of Colombia.
Overall, I think this manuscript should consider major revisions, from revising the introduction, improving the clarity of the concluding statements, to the presentation of the figures. I have stopped reading in-depth in section 3.1 because I found the manuscript to be hard to follow and the authors should try to sharpen the focus on what they are presenting. I think the presentation and explanation of how they use the methods and the context of its use in their experiment should be significantly improved. I would encourage major revisions and resubmission after carefully rewriting.
However, the paper is hardly readable; it is hard to link the findings to the methods. I am raising several concerns in the review Pdf file.
Major:
- The English needs to be improved dramatically. Obvious grammar mistakes can be found frequently and the writing style is poor.
- I find the abstract to be unfocused, bringing up several topics and switching between the topics without a clear message. I would suggest to revise the abstract to be more concise on what the authors have done and their main results.
- I think the Introduction is very drawn-out and at points a bit confusing on the direction of the story. I think the Introduction should be revised to follow this structure:
What is the problem
Why is it important
What was done so far
What is left to do
What the authors plan to do and please include hypothesize if possible. It will improve the readership.
Minor:
- By reading your paper, I think the survey design is particularly challenging, therefore detailed description is required. Adding lots of adjectives like representative, valid, accurate or reliable does not mean you have done something extraordinary. A more object way of describing your method is appreciated.
- I am having issues following the exact procedures of the sample selections.
- The results are difficult to follow, and I cannot see how they are linked to the analytical methods. I suggest to carefully pick the statistical methods and describe them in details. Meanwhile please describe them in a non-technical way to ensure that the readers won't feel overwhelmed.

Author Response
February 7 2022
Dear Assistant Editor and Reviewers
Please find attached a revised version of our manuscript agriculture-1532735 entitled “Nature’s contributions to people shape sense of place in the Coffee Cultural Landscape of Colombia” to be considered for publication in Agriculture Special Issue. Overall, this revised version has been substantially edited and reworded and incorporates all major and minor suggestions provided by both reviewers.
Overall, major revisions in this revised version are summarized below:
- We have substantially reworded the abstract, introduction, and discussion according to specific recommendations provided by R1 and R2.
- Key figures of this research have been edited, and new information is now incorporated in new Appendices to clarify concerns regarding methods and material
- Bibliography has been extensively updated based on reviewers suggestions
- This revised version has been edited by a native English speaker
For a clearer understanding, a table has been created with a response to each reviewer comment (please see the attachment). We truly want to thank to both editor and reviewers for providing positive insights and helpful comments to improve our manuscript.
Sincerely,
Beatriz Elena Murillo-López and Antonio J. Castro, on behalf of the authors

Reviewer 2 Report
In this paper the authors use a mixed methods approach (qualitative and quantitative) to describe the values, in particular sense of place, of farmers in a key UNESCO farming region. Their data is rich and compelling. However, some work is needed to frame the questions proposed by the authors to recent literature and have the data and codes reflect conversations about of sense of place and relational values. Likewise, it would be helpful to describe the source of the codes used in classifying the semi-structured interviews. These were not terms I am familiar with (I don’t mean to suggest they are not correct, but they some context is needed). I also think it would be helpful to be clearer in the description of the land use and land cover and how this is important to the conclusions drawn from the interviews. It seems very important to the framing, but terms are used loosely (specifics noted below) but this includes being clear on grain, extent and definition. Most importantly as noted above, it think the authors could improve the impact of their work by demonstrating how it is an extension of other work on farmer values in agricultural landscapes.
Specific responses
Line 50 – Perhaps a bit of nuance could be added to differentiate between “natural systems” and “traditional farming systems” given that if land is cultivated it starts to move away from natural quickly. Likewise, in line 67, the threat of agricultural intensification is noted. Could this be discussed here as well?
Line 77 – this seems like an unsupported statement. I think a stronger review of SoP (particularly in agricultural landscapes) would be helpful.
Line 89 – Likewise perhaps consider the relational values work of Allen et al specific to relational values in agricultural systems
Methods
Line 153 – is the focus on Land Cover change as above, if so the intro needs to be stronger, or the intro needs to be better connected to the methods
Lines 201-216 on farm styles is not clear. I see after reading to “styles” in the results but this process is not clear
Line 240-241 – this list seems more like results. That being said, I can see how they might blur given the context of the research.
Line 257 seems to be unedited, I think I know what is being asked, but please double check and clarify.
Results –
Line 307 – this estimate at 33% seems to contradict the framing in the introduction of industrial as the norm. Likewise, the treat of urbanization is not captured here.
Line 322,what is in the third dimension?
Section 3.2 (lines 335-345) – where did these codes come from? Are these connected back to relational values? NCP? The process or source of these classifications should be in the methods. How were these quotes chosen?
Line 349- related to the frequency of reported? Is this per farm? Or could one farmer report this multiple times?
Lines 364-366. This is super interesting! Would love to see more around this with quotes. Indeed, the quotes are a rich source of data. I would love to see these better organized and emphasized
Lines 363-383 – Most of these do not seem to be about sense of place.. Perhaps a better review of the SoP literature can help address this.
Author Response
February 7 2022
Dear Assistant Editor and Reviewers,
Please find attached a revised version of our manuscript agriculture-1532735 entitled “Nature’s contributions to people shape sense of place in the Coffee Cultural Landscape of Colombia” to be considered for publication in Agriculture Special Issue. Overall, this revised version has been substantially edited and reworded and incorporates all major and minor suggestions provided by both reviewers.
Overall, major revisions in this revised version are summarized below:
- We have substantially reworded the abstract, introduction, and discussion according to specific recommendations provided by R 1 and R2.
- Key figures of this research have been edited, and new information is now incorporated in new Appendices to clarify concerns regarding methods and material
- Bibliography has been extensively updated based on reviewers suggestions
- This revised version has been edited by a native English speaker
For a clearer understanding, a table has been created with a response to each reviewer comment (please see the attachment). We truly want to thank to both editor and reviewers for providing positive insights and helpful comments to improve our manuscript.
Sincerely,
Beatriz Elena Murillo-López and Antonio J. Castro, on behalf of the authors
